# Clinical Evaluation of Different Treatment Strategies for Motor Recovery in Poststroke Rehabilitation during the First 90 Days

**DOI:** 10.3390/jcm10163718

**Published:** 2021-08-21

**Authors:** Ekaterina S. Koroleva, Stanislav D. Kazakov, Ivan V. Tolmachev, Anton J. M. Loonen, Svetlana A. Ivanova, Valentina M. Alifirova

**Affiliations:** 1Department of Neurology and Neurosurgery, Siberian State Medical University, 2 Moskovsky Trakt, 634050 Tomsk, Russia; kattorina@list.ru (E.S.K.); ivanovaniipz@gmail.com (S.A.I.); v_alifirova@mail.ru (V.M.A.); 2Department of Medical and Biological Cybernetics, Siberian State Medical University, 2 Moskovsky Trakt, 634050 Tomsk, Russia; ivantolm@mail.ru; 3Unit of PharmacoTherapy, Epidemiology & Economics, Groningen Research Institute of Pharmacy, University of Groningen, Antonius Deusinglaan 1, 9713AV Groningen, The Netherlands; a.j.m.loonen@rug.nl; 4Department of Psychiatry, Addictology and Psychotherapy, Siberian State Medical University, 2 Moskovsky Trakt, 634050 Tomsk, Russia

**Keywords:** ischemic stroke, augmented reality rehabilitation, motor recovery, movement assessment, early recovery period, clinical and functional outcomes

## Abstract

Background: Motor recovery after stroke is based on neuronal plasticity and the structural reorganization of the brain. Questions are debated about the proper moment to start rehabilitation in the acute period of stroke, the significance of rehabilitation interventions during the so-called “plastic window”, and the advantages of modern and traditional programs. The aims of this study were to evaluate the role of different rehabilitation strategies and their combinations for motor recovery and the impact on functional disability by way of neurological and functional outcomes 3 months after ischemic stroke. Methods: We used three rehabilitation approaches: early rehabilitation from the first day of stroke (Phase I), traditional exercise programs (Phase II), and an author’s new method of biofeedback rehabilitation using motion sensors and augmented reality (AR) rehabilitation (Phase III). Clinical and functional outcomes were measured on the 90th day after stroke. We developed algorithms for quantifying the quality of movements during the execution of tasks in the motor domains of the AR rehabilitation program. Results: Phase I of rehabilitation led to an improvement in functional independence, and the recovery of motor functions of the extremities with an absence of mortality and clinical deterioration. AR rehabilitation led to significant improvement both with respect to clinical and functional scores on scales and to variables reflecting the quality of movements. Patients who were actively treated during Phases II and III achieved the same final level of motor recovery and functional outcomes as that of participants who had only received AR rehabilitation during Phase III. Patients who underwent outpatient observation after Phase I showed a deficit of spontaneous motor recovery on the 90th day after stroke. Conclusions: Early rehabilitation was successful but was not enough; rehabilitation programs should be carried out throughout the entire “sensitive period” of poststroke plasticity. The newly developed AR biofeedback motion training is effective and safe as a separate rehabilitation method in the early recovery period of moderately severe, hemiparalytic, and ischemic stroke. These two rehabilitation approaches must be applied together or after each other, not instead of each other, as shown in clinical practice.

## 1. Introduction 

Stroke continues to be a topical problem. In most countries, stroke is the leading cause of death and one of the main reasons for adult-onset disability [1]. In 2013, the number of stroke survivors throughout the world was estimated to be 25.7 million, and this number is calculated to reach 70 million by 2030 [2,3]. Motor deficit is the most common health-related problem caused by stroke [1,4]. After stroke, about 85% of patients experience motor impairment that affects their activities of daily living (ADLs) and quality of life [5,6]. These figures illustrate why stroke patients need effective rehabilitation programs to achieve the highest possible level of physical independence.

Brain recovery from stroke is based on neuronal plasticity that is generally defined as changes or rewiring in the neural network. Most stroke deficits see the highest rate of recovery during the first 3 to 6 months after stroke because, during this period of time, the brain is most susceptible to restoration [7]. There are two key factors that promote neuroplasticity during the above-mentioned “plastic window”. First, ischemic damage to the brain tissue activates angiogenesis and the sprouting of new dendrites and axons, which is probably related to the enhanced expression of neuronal growth factors during the acute period of stroke [8]. This mechanism presumably underlies the phenomenon of spontaneous recovery. Second, rehabilitation therapy during the early poststroke period may induce plasticity and improve recovery outcomes. Although recent preclinical work demonstrated the positive effects of rehabilitation to reduce motor impairment, similar data do not exist in humans [9,10]. This led to the question: could rehabilitation interventions in humans amplify recovery? One potential solution is to find the optimal timing and dosage of current interventions that employ motor training and mobilization so as to augment what is expected from spontaneous recovery [11]. Therefore, providing clinical studies aimed at solving this problem is of great importance for neurorehabilitation. 

Several studies have shown the potential of early rehabilitation to improve motor recovery after stroke [8,12]. However, harmful effects of early rehabilitation were simultaneously demonstrated, such as increasing stroke volume and the risk of hemorrhagic conversion [13]. Hence, evidence supporting the advantages of rehabilitation interventions during the early stroke period is still inconclusive. The same is true for the duration of the sensitive period considering the beneficial and harmful effects. It was estimated that the first period probably lasts for 1–3 months because, during this time period, most recovery from impairment occurs [11]. The time interval during which the patient is particularly sensitive to additional damage can be expected to be far shorter.

Several rehabilitation approaches are used to increase the restoration of motor deficit and the improvement in quality of life [1,4,5]. The standard care for stroke includes a number of traditional motor rehabilitation techniques, and aerobic exercise is its inherent part. Aerobic exercise promotes motor unit recruitment, the improvement of functional capacity, and greater confidence in engaging in physical activities [11]. Currently, modern methods such as robot-mediated therapies (RMT) and virtual reality (VR) rehabilitation are applied in addition to traditional rehabilitation therapy. A number of robotic devices have been proposed for use in clinical practice. Robotics provide motion assistance and aid with patient movement. Nonetheless, no convincing data on their positive impact on true recovery from stroke were found, and clinical research in this field continues [14]. Robotic devices most likely promote neuroplasticity by the systematic and intensive repetition of specific movements, which is crucial for the motor-learning process [5,15]. Positive effects on motor recovery were also shown to result from applying neurorehabilitation involving VR technology. VR therapy may induce brain plasticity by thorough sensorimotor cortex activation. The benefits of VR rehabilitation techniques include task-specific practice, and visual and auditory biofeedback applied in an entertaining environment [16,17].

Despite the recent developments in neurorehabilitation techniques focusing on motor recovery after stroke, no universally accepted model of poststroke rehabilitation exists, and research on optimal therapeutic interventions and the optimal time period to apply them remains pertinent [5,18]. The authors developed a new rehabilitation technique, termed motor rehabilitation, using motion sensors and augmented reality (AR). The aims of this study were to evaluate the role of different rehabilitation strategies and their combinations for motor recovery and the impact on functional disability by way of neurological and functional outcomes 3 months after ischemic stroke.

## 2. Material and Methods 

### 2.1. Patient Recruitment

The study was conducted by the Department of Neurology and Neurosurgery in collaboration with the Department of Medical and Biological Cybernetics of the Siberian State Medical University (SSMU), Tomsk Regional Vascular Center (Tomsk RSC), Siberian Federal Scientific Clinical Center of Federal Medicobiological Agency, Russia from November 2018 to December 2019. The study was conducted in accordance with Resolution 5961/2018. The study protocol was reviewed and approved by the Medical Research Ethics Committee of SSMU, Tomsk, Russia (protocol no. 5961, 18 June 2018). All participants provided informed consent prior to participation.

### 2.2. Patient Characteristics

Fifty stroke patients aged 18 years or more were recruited from the Tomsk RSC within the first 24 h after the onset of symptoms. Ischemic stroke in the middle cerebral artery basin was diagnosed considering their clinical history, the presence of focal neurological signs and/or symptoms, and by applying magnetic resonance imaging (MRI) or computer tomography (CT) scans. Additional inclusion criteria were Glasgow Coma Scale (GCS) of no less than 15 points, and Mini-Mental State Examination (MMSE) of no less than 20 points [19,20,21]. Thus, this study was carried out in patients with moderate and mild stroke. Stroke patients with severe disabilities, or people with severe cognitive or communication problems were excluded. In addition, patients with subarachnoid, extradural, or subdural hemorrhage; transient ischemic attack; history of prior stroke; neurological deficits related to trauma, neoplasm, or psychiatric disorders; excessive use of alcohol or drugs were excluded. Patients were also excluded who had received thrombolysis and thrombectomy, who had developed hemorrhagic transformation, or who were clinically unstable. 

### 2.3. Study Design

The neurorehabilitation of stroke patients in early recovery period (from the 1st to the 90th day of stroke) was carried out in 3 phases according to the current legislation of the Russian Federation.

Phase I—early rehabilitation within the first 2 weeks of acute stroke was mandatory for all patients from the 1st day of admission to Tomsk RSC according to Order of the Ministry of Health of the Russian Federation no. 928n, “On approval of the procedure of care for patients with acute disorders of cerebral circulation”, dated 15 November 2012 [22]. The time range of early rehabilitation was 14–16 days. The complex of measures for early rehabilitation was carried out by a multidisciplinary team of specialists in accordance with Order of the Ministry of Health of the Russian Federation no. 1740n dated 29 December 2012, “On approval of the standard of specialized medical care for brain infarction” [23]. Motor rehabilitation included the use of passive–active cyclic robotic electromechanical technologies (Armeo Power Hocoma robotic arm exoskeleton, MOTOmed viva2 simulator). Early rehabilitation treatment was applied from the 1st day of stroke in patients with hemodynamic stability. 

Phase II—inpatient rehabilitation within 28 days of stroke in Siberian Federal Scientific Clinical Center of Federal Medicobiological Agency in accordance with Order of the Department of Health of the Tomsk region no. 941, “On medical rehabilitation in specialized medical organizations at the inpatient stage”, dated 15 March 2018 [24]. The approved procedure for the organization of medical care establishes the rule of referral to the hospital of persons with a neurological profile over the age of 18 years after receiving specialized medical care in the Tomsk RSC [25]. The primary aim of Phase II was restoring the mobility of the patient by traditional exercise programs and neuropsychological approaches. The rehabilitation program and its modes were individually selected per each patient. Details of the applied traditional rehabilitation techniques are provided in Appendix A. During rehabilitation, the patient’s functional state was monitored by assessing heart rate, blood pressure, and blood-oxygen saturation.

Phase III—outpatient rehabilitation within 12 days of stroke in SSMU in accordance with Order of the Department of Health of the Tomsk region no. 188, “On medical rehabilitation in outpatient stage”, dated 16 October 2018. Neurological patients who did not need round-the-clock monitoring and the individual care of junior medical staff were sent for outpatient rehabilitation [26]. Motor rehabilitation used motion sensors and augmented reality (AR). AR rehabilitation was conducted in the SSMU with specialized software developed by us that aimed at motor rehabilitation using motion sensors and augmented reality (NeuroRAR software. Certificate of state registration of a computer program no. 2019619367, dated 16 July 2019).
Observation period: within 90 days of stroke.Observation points were selected according to phases of rehabilitation (Figure 1):
Point 1—first day of stroke;Point 2—after Phase I of rehabilitation (median 14th day, range 14–16);Point 3—after Phase II of rehabilitation (median 45th day, range 16–60);Point 4—after Phase III of rehabilitation (median 82th day, range 60–90).

The study population was clinically and demographically homogeneous on admission to the RSC at Point 1. Baseline characteristics are specified in Table 1. After Phase I, which was mandatory for each patient, the population was divided into three groups: Groups A–C. 

Group A: 21 patients who received restorative treatment during consecutive rehabilitation Phases I–III within 90 days of stroke.Group B: 14 patients who received treatment during Phases I and III. Group B patients refused inpatient rehabilitation on Phase II and, during this time period (range 14–60 days), were on outpatient observation.Group C: 15 patients who only received treatment during Phase I. Group C patients refused further rehabilitation and were followed up on outpatient observation and examined on the 90th day of stroke.

The absence of treatment in Phases II and III of rehabilitation in Group C and in Phase II of rehabilitation in Group B was associated with problems of routing and optimizing Phases II and III of the rehabilitation process for cerebral stroke, and problems in the family. Most of these patients lived in geographically remote areas of the Tomsk region.

### 2.4. Applied AR Rehabilitation Techniques

In our research, we used the augmented reality (AR) system. The effectiveness of neurorehabilitation largely depends on the motivation for correct exercise performance. In the case of AR, the accuracy and speed of task execution directly affects the patient’s assessment of their own capabilities; in this case, the achievement of a certain result motivates the patient to further perform the tasks. An important aspect is also comfortable orientation in the surrounding space without additional stress factors caused by the isolation of the visual analyzer, as happens in VR conditions. When compiling the tasks, we focused on restoring upper limb function: 3 out of 4 tasks were related to hand-control skills. We also applied the method for assessing the spectral power of the trajectory of hand movement described by Balasubramanian [27].

To visualize the virtual environment, we used Epson Moverio BT-300 augmented reality glasses with preinstalled mobile module software NeuroRAR. For biofeedback, we used Markerless motion-capture systems Leap Motion and Microsoft Kinect. The involved kinesiotherapist could control AR rehabilitation using desktop module software NeuroRAR.

This software implements four different motor domains for neurorehabilitation:(a)Accuracy domain (Figure 2a). The patient performs the task in a sitting position. The scene represents cubes of two colors (red and black) with selective numbering. The patient alternately clicks on the cubes, performing extensor movements in the shoulder and elbow joints. Each time after pressing, the patient returns the upper limb to its original physiological start position. The exercise is aimed to train the muscles of the shoulder girdle and upper limb, increasing the strength and accuracy of movements.(b)Statics domain (Figure 2b). The patient performs the task in a sitting position. The virtual scene represents a circle with a triangle inscribed within it. The patient holds the index and middle fingers folded together against the contour of the circle when executing the task, and then moves their arm by the contour of the circle and triangle clockwise and counterclockwise. The virtual objects change color from red to green when the patient succeeds. This task has a high level of difficulty, allowing for training the reciprocal interaction of muscles with a static–dynamic load. An example of how the task is executed is presented in Figure 3a,b.Several parameters reflecting the execution of this task can be applied to measure the quality of the made movements. Augmentation of the “variability of movements” parameter can be interpreted as making a smoother movement during the task, and by decreasing the severity of intentional tremor during movements. This variable increases when the forefinger is moved closer to the main trajectory without making extra movements such as tremors and sudden movements to the side. Increasing the “number of completed tasks” during one motor training session and enhancing the “maximal duration of movement in one approach” is gradual improvement in the control of reciprocal muscles reflected by an increase in the duration and speed of the task.(c)Capture domain (Figure 2c). The patient performs the task in a sitting position. The virtual scene represents numbered red or black balls. The patient alternately captures and compresses the balls, first the red and then the black. The exercise was designed to train hand muscles and develop finger functionality by the formation of the correct grip.(d)Balance domain (Figure 2d). The patient performs the task in a standing position while walking on the spot. The virtual scene represents a yellow road moving under the patient. The patient takes a step from time to time over a virtual obstacle and grabs objects flying past them. The exercise attempts to train the correct walking skill, maintain balance, and overcome obstacles when walking. An example of how the task is executed is presented in Figure 3c,d.

Increasing the “height of raising the paretic leg” was evaluated relative to the set level. The level of leg raising was assessed using a height of 0.25 m for the approaching virtual barrier on a flat horizontal surface. The “variance of the displacement for the central point” variable shows displacements of the point corresponding to the geometric center of the body. This variable shows how much a person sways in the process of a calm walk between stepping over obstacles. Indirectly, this parameter can characterize the state of human balance when performing routine movement.

The AR rehabilitation program includes 10 daily motor training sessions (except weekends) with a frequency of 1 motor training session per day according to AR protocol. The duration of one motor training session was 60 min. The time to complete one motor domain was 15 min. The kinesiotherapist assisted the patients and taught them how to execute the AR tasks in the neurological clinic.

### 2.5. Clinical Assessments

Sociodemographic and clinical data were extracted from electronic medical records and/or by interviewing the patients. The evaluation of neurological impairment was based on the National Institutes of Health Stroke Scale (NIHSS) [28,29]. The Fugl-Meyer Assessment (FMA) scale was used as a standardized test for the comprehensive evaluation of motor function in hemiparesis poststroke recovery [30,31]. Motor control of the patients’ paretic arm was assessed by the upper extremity section of the Fugl-Meyer scale (FMA-UE) and of the paretic leg with the low extremity section [32,33].Functional disability and ADLs were assessed on a modified Rankin Scale (mRS) [34]. We used a modified Ashworth scale (MAS) for the quantitative evaluation of upper limb spasticity [16,35]. Etiologic subtypes of ischemic stroke were classified according to the Trial of ORG 10172 in Acute Stroke Treatment (TOAST) criteria [36]. Clinical data were registered (clinical, paraclinical, and neurobiological characteristics of patients with ischemic stroke at phases of medical rehabilitation from day 1 to 90 of acute cerebrovascular attack. Certificate of state registration of a database no. 2020622069, dated 27 October 2020).

### 2.6. Statistical Analysis

Data were analyzed with the IBM SPSS Statistics for Windows ( IBM Corp. Released 2011., Version 20.0. Armonk, NY, USA). To calculate the sample size, we also used specialized software complex IBM SPSS Sample power. In our study, there were three groups. The key parameters were connected with clinical assessment (FMA, NIHSS), and we used these variables to calculate the sample size. For 80% power, the minimal sample size was 13. Results are expressed as median (Me) and interquartile ranges (IQR); for categorical variables, results are expressed as percentages. The Kolmogorov–Smirnov test was used to analyze the normal distribution of the variables (*p* > 0.05). Quantitative data without normal distribution were analyzed with nonparametric tests. Differences in the parameters between groups were assessed with the nonparametric Mann–Whitney U test; for related samples, we used the Wilcoxon rank-sum test. The chi-squared (χ^2^) test was used to assess differences in the size of distribution of qualitative data. The Bonferroni test was used as a lead for multiple post hoc comparisons. Statistical analysis was conducted on the 95% confidence level. A *p* value less than 0.05 was considered statistically significant. 

## 3. Results

No significant differences at baseline were found between Groups A–C in terms of age, sex, or vascular risk factors (Table 1). 

### 3.1. Assessment of Clinical and Functional Measures

At entry to Tomsk RSC, no differences existed between patients of the three groups with regard to stroke severity as measured by NIHSS and FMA, and in overall functional performance as measured by mRS at Point 1 (*p*_A–B_; *p*_A–C_; *p*_B–C_ in Table 2. Neurological deficits in the acute period of ischemic stroke were assessed as being of moderate severity. However, at other points, the score on the clinical and functional scales differed among patient groups who had received different rehabilitation programs.

Table 3 shows the detailed dynamics of changes in the clinical and functional assessment scales in patient groups during the observation period.

Patients in Group A showed a linear improvement in scorings according to the assessment scales as a result of using consecutive rehabilitation programs (Phases I–III). Significant difference was found between phases of rehabilitation by NIHSS, mRS, and total FMA (*p*_1–2_; *p*_2–3_; *p*_3–4_). Detailed comparative analysis of the FMA showed that motor impairment of the upper and lower extremities (FMA-UE and FMA-LE) regressed stepwise after each rehabilitation program, with statistical differences between observation points (*p*_1–2_; *p*_2–3_; p_3–4_). There was no statistical difference between FMA-Balance-2 and FMA-Balance-3 scores (*p*_2–3_ = 0.06).

Comparisons of the scores (NIHSS, mRS, and total FMA) after Phases I and III of rehabilitation in Group B demonstrated equally significant regression of motor impairment (p_1–2_; p_3–4_). Apart from NIHSS ratings, they did not significantly change between Points 2 and 3 in patients of Group B, who did not receive a traditional rehabilitation program (Phase II) and underwent outpatient observation during this time (*p*_2–3_). Scores on the NIHSS scale were significantly lower at Point 3 (*p*_2–3_ = 0.010). Group B showed no change in motor impairment by total FMA scale during the period of outpatient observation (p_2–3_). The situation was similar with regard to ADLs measured by mRS (*p*_2–3_). Evaluation of lower limb motor function by FMA-LE in Group B patients demonstrated improvement after Phase I and a significant difference between Points 2 and 4 (*p*_1–2_; *p*_2–4_). However, no significant differences were observed between FMA-LE-2 and FMA-LE-3 (period of outpatient observation), and between FMA-LE-3 and FMA-LE-4 (AR rehabilitation). There was an absence of significant recovery as assessed by FMA-LE due to the stimulation of the motor cortex by the AR biofeedback motion training (*p*_3–4_). 

The clinical condition of patients in Group C was assessed after Phase I of rehabilitation and on Day 90 poststroke. Significant clinical and functional improvements were shown after early rehabilitation in Tomsk RSC in Group C patients (*p*_1–2_). The rating at 90 days poststroke showed that their clinical condition did not significantly change during outpatient observation, in which they had not received active treatment (*p*_2–4_).

Differences between the three groups of patients are shown in Table 3. Neither on the 1st nor the 14th day poststroke was any significant difference measured with NIHSS or mRS between the three groups (*p*_A–B_; *p*_A–C_; *p*_B–C_). However, Group A patients had significantly lower total FMA-2 scores than those of Group B patients after Phase I of rehabilitation (*p*_A–B_ = 0.01). Detailed comparative analysis of the FMA subscores showed that the lower scoring of patients of Group A was due to selective bad recovery of the upper extremities after early rehabilitation in comparison to Group B. However, patients in Groups A and B showed no significant difference between the total and subscale FMA-3 scores, and no significant differences were observed with respect to NIHSS-3 and mRS-3. This was in spite of Group A patients receiving active treatment/a traditional rehabilitation program (Phase II), and Group B patients being on outpatient observation during this time period (*p*_A–B_). Comparisons of the scores after AR rehabilitation (NIHSS-4, FMA-4, and mRS-4) demonstrated an equally significant regression in motor impairment both in the paretic arm and in the paretic leg in Group A and B patients (*p*_A–B_). Therefore, despite the fact that Group A patients had lower FMA-UE-2 scores, at the end of the study they achieved the same level of upper limb motor recovery, assessed by FMA-UE-2, as that in Group B.

Functional outcomes after three months, as assessed by mRS-4, and for neurological deficits and outcomes of motor recovery, as assessed by NIHSS-4 and FMA-4, showed significantly poorer results in Group C patients (*p*_A–C_; *p*_B–C_). Patients in Group C, who were admitted to outpatient observation directly after Phase I of rehabilitation, had worse clinical parameters and ADLs in comparison to patients in Groups A and B at the end of the "plastic window" period. However, some special features of recovery were evident. Motor function of the lower limbs in patients in Group C recovered as well as in patients in Group A by the 90th day poststroke, and there were no significant differences in FMA-LE-4 (*p*_A–C_). The balance function was also not statistically different with regard to the FMA-Balance-4 score between the groups (*p*_A–B_; *p*_A–C_; *p*_B–C_). 

### 3.2. Spasticity Assessment 

We measured upper limb spasticity with the modified Ashworth scale (MAS). Directly after stroke, the MAS-1 score was equal to 0 in patients of Groups A– C. At discharge from Tomsk RSC after two weeks of early rehabilitation, spasticity was registered by MAS-2 as +1 in two patients of Group A (10%) and three patients of Group C (20%). However, muscle tone in Group A did not differ during phases of rehabilitation MAS-2 = MAS-3 = MAS-4 = +1. In the same time period, patients of Group C showed an increase in spasticity on the 90th day poststroke (MAS-4 = 3). Group B patients did not have spasticity. 

### 3.3. Assessment of Movement Quality during AR Rehabilitation

Comparison results of the quality of movements in Group A and B patients are shown in Table 4 and Table 5. We used two tasks to assess the quality of motion during AR rehabilitation: statics and balance. The statics task was aimed at developing the reciprocal interaction of upper limb muscles with a static–dynamic load (Figure 2b and Figure 3a,b). The balance task addressed lower limb function and balance during walking (Figure 2d and Figure 3c,d). Comparison of the measures during the first, fifth, and tenth sessions did not produce significant differences between the two groups of patients for any of the calculated parameters, with one exception: considering the maximal duration in one approach, we observed that Group A patients repeated the task during the first session significantly slower than Group B patients (Table 4). Measurement of this parameter yielded no significant difference between Group A and B patients during the fifth and tenth training sessions (Table 4). Within-group comparisons resulted in a similar pattern of motor quality improvement of the patients in both groups (Table 5).

We observed a significant increase in the “variability of movements” parameter of the statics domain in both patient groups (Table 5). The accuracy of following a given trajectory increased during the process of motor training (Figure 3a,b). We observed a steady progression of this criterion from the first to the tenth session of AR rehabilitation in patients of Groups A and B. We also found a significant increase in the variables of “number of completed tasks during one motor training session” and “maximal duration of movement in one approach” while executing this statics task. This last variable was predominantly enhanced between the first and fifth sessions, while the “number of completed tasks” parameter was augmented more gradually between the first and last sessions. These changes reflected an improvement in the control of reciprocal muscles while executing the task. 

We observed that, within the balance domain, patients significantly increased the height of raising their leg on the affected side during the fifth and tenth sessions of AR rehabilitation in comparison to during the first session (Table 5). By the fifth session, patients had already coped with this task, and the affected leg was not lifted higher by the 10th visit in either Group A or B patients. At the same time, only the “variance of the displacement for the central point” variable significantly decreased from the fifth to the tenth session of AR rehabilitation in both groups. 

## 4. Discussion

In this study, we assessed the clinical and functional effects of different rehabilitation programs applied to different combinations during the “plastic window” period in stroke patients. All patients included in our study received motor rehabilitation with early activation during Phase I of the program (i.e., the period of the first 14 days after stroke). Today, early stroke rehabilitation is recommended in many clinical practice guidelines, including the Order of the Ministry of Health of the Russian Federation no. 928n [22,37], but the results of clinical studies on the effects of motor rehabilitation in the early recovery period of stroke are conflicting. According to the data from some clinical trials, very early rehabilitation within the first 24 h of stroke may be harmful [8]. Meta-analysis of nine randomized controlled trials with 2958 participants, including one of 2104 participants, showed that very early mobilization (VEM), which involves the first mobilization within 24 h of stroke onset, may be accompanied with higher mortality and dependency, and no better treatment results compared to delayed mobilization [38]. From the perspective that the risk of a negative effect of very early mobilization probably depends on stroke subtype, stroke severity, and receiving treatment with tissue plasminogen activator (tPA), we only included moderately severe ischemic stroke patients without tPA treatment. Evidence of the effective use of early exercise after stroke in humans is currently limited by the lack of data [8,12,39]. The influence of early exercise on postischemic rehabilitation has mostly been demonstrated in preclinical studies using experimental animal models [40,41]. The beneficial effect of early rehabilitation has been connected with exercise intensity, initiation time, and exercise program [42]. There exists no "gold-standard" time for the start of rehabilitation, and science has not yet set clinically relevant sensitive periods after stroke. 

The results of our study demonstrated that early rehabilitation, starting within the first 24 h of stroke, was successful. The primary outcome was improvements in functional independence as measured by mRS, and recovery of motor functions of the upper and lower extremities measured by FMA. Patients in each of the three groups showed significant changes in the evaluation scales between Points 1 and 2. It is difficult to estimate which part of motor recovery was due to spontaneous biological recovery based on structural neuroplasticity, and how much was dependent on the rehabilitation treatment based on synaptic plasticity [4,8]. Upper limb motor function in the patients of Group A had a worse recovery after early rehabilitation than those in Group B measured by FMA-2. The study population was clinically and demographically homogeneous, all patients had the same rehabilitation potential, evaluated at admission to Tomsk RSC, early rehabilitation was performed according to the standards in accordance with the Order of the Ministry of Health of the Russian Federation no. 1740n [23]. Diversity in innate plasticity, and the molecular and cellular mechanisms of the pathophysiological processes involved in motor impairment are possible reasons for this discrepancy between Groups A and B [8]. Secondary safety outcomes were the absence of mortality and intracranial hemorrhages in patients who had received early rehabilitation. The intensity of rehabilitation was moderate, and the modes of early rehabilitation were selected individually for each patient. The professionals most likely applied additional predictors of successful recovery, helping to avoid complications and clinical deterioration. Specific clinical factors of patients with acute ischemic stroke affect not only the outcome of early rehabilitation but are also considered when using other treatment approaches for acute ischemic stroke. The recent prospective cohort study showed that intra-arterial mechanical thrombectomy, either in combination with intravenous thrombolysis or alone in patients with acute ischemic stroke, can be associated with good outcomes at 3 months [43]. The determinant for 90-day measurement was a baseline NIHSS score. It was also shown that the reperfusion seems to be not the only determinant of good outcomes. It can be supposed that some aspects of patient management in the intensive care unit, the individual therapeutic and rehabilitation approaches in the acute phase of ischemic stroke, can counterbalance the positive effects of reperfusion and further improve neurological outcomes.

The sensitive period of poststroke plasticity, when maximal recovery from neurological lesions is possible, is defined as the first 1–3 months after stroke [44,45]. This is why neurorehabilitation in the early recovery period of a stroke is so important. During this period, spontaneous and treatment-related neuroplastic changes allow for the recovery of movement patterns back to the pre-stroke level. During later periods of stroke, improved motor function is more often associated with compensatory responses, alternative movements, and adaptive strategies during the execution of motor tasks. The cognitive functions of stroke patients are directly related to their involvement in the rehabilitation process, and they may affect the results of recovery [46,47]. Therefore, one of the criteria for inclusion in our study was testing on the MMSE scale and the absence of a pronounced cognitive deficit. However, stroke patients often have socioeconomic and family problems, as well as the problem of geographical distance from rehabilitation centers. Patients of Group C received only early rehabilitation during the 90 day study, and patients of Groups A and B showed clinical and functional improvements after Phase I. Conditions in Group C patients did not significantly change during outpatient observation according to statistical data (*p*_2–4_). However, they showed the worst clinical and functional outcomes by NIHSS, mRS, total FMA, and FMA-UE at the 90th day after stroke onset compared to those of Groups A and B. This indicates the negative impact of the absence of physical activity and rehabilitation interventions for the recovery of movement patterns in the upper extremities and ADL. At the same time, Group C patients showed no spontaneous recovery in the motor function of paralytic legs according to FMA-LE-4, and there were no significant differences between patients of Groups C and A. We also found no significant differences in FMA-Balance-4 between Group C and Groups A and B (*p*_A–C_; *p*_B–C_). With regard to the balance function, we could not exclude the role of compensatory adaptive mechanisms. In addition, there is the opinion that the rates of motor recovery for upper and lower extremities are different [48]. Paretic-arm recovery is more complicated compared to that of lower limbs. Thus, this hypothesis could explain data resulting from this study. 

In our study, we were able to assess the effects of classical stroke rehabilitation treatment for four weeks (Phase II) and the effects of applying a developed AR rehabilitation program (Phase III), and to compare the results of motor recovery in patients in Group A who had received Phase II and III treatment with patients in Group B after the AR rehabilitation program who underwent outpatient observation during Phase II. Clinical and functional outcome measures after a 90-day poststroke period showed no significant difference in Groups A and B. Members of Group A achieved the same level of motor recovery and functional outcomes after finishing the traditional rehabilitation program (Phase II) as that of Group B participants who underwent outpatient observation in this time period (Point 3). In Group A patients, motor control of the paretic arm, as assessed by FMA-UE-2, was significantly worse after Phase I in comparison to that of Group B, and significantly improved after Phase II, without significant differences in FMA-UE-3 between groups (*p*_A–B_). So, Phase II of motor rehabilitation with classical recovery approaches showed significant motor improvement in patients of Group A who showed no significant difference from participants of Group B by the clinical and functional scales at Point 3.

Patients in Group B who had not received active treatment during Phase II of the study did not show any improvement in motor impairment by FMA and mRS scores at Point 3. This showed the deficiency of spontaneous motor recovery during the outpatient observation period. The average NIHSS score was statistically lower (*p*_2–3_). This can also be considered as an element of spontaneous-recovery manifestation. However, the NIHSS scale does not only characterize motor impairment, and the number of points could be decreased by restoring other impaired functions (speech, sensory disorders).

AR rehabilitation led to significant improvement both with respect to the scores on scales measuring clinical and functional performance, and to variables reflecting the quality of movements in patients of Groups A and B. During the first session of AR rehabilitation, patients who had received active treatment during Phase II performed similarly (worse concerning a single parameter) in comparison to patients who had not. Patients of Group C who had not received active treatment performed significantly worse after three months and showed hardly any improvement after the first phase of the program. This may indicate the effectiveness of motor rehabilitation during Phases II and III of the program.

The beneficial effects of the AR rehabilitation program are reflected by significant improvement during Phase III according to clinical and functional performance, and quality of movement measures in patients of Groups A and B. Interpreting the results, it becomes clear that muscle strength in the affected arm of patients in Groups A and B increased between the first and tenth AR biofeedback motion training sessions. Generally speaking, the movement accuracy and muscle force of the affected arm were augmented in Group A and B patients (see example of trajectory in Figure 3a,b). The obtained results by movement quality assessment during AR rehabilitation corresponded to the related FMA score. Results concerning the movement of the affected leg are somewhat more difficult to interpret because lower-extremity movements determine walking and maintaining stability in an upright position. The affected leg (in a rigid flexed position) is more stably supported and, due to this stability, fewer deviations of the trunk occur. We observed that the height of raising the legs during walking on the spot on a flat horizontal surface was lower during the first days of training, which in turn led to a decrease in the dispersion of movement of the central point of the body. This observation indicates that, during the training process of walking on the spot, patients could maintain their balance better and swayed less. In this case, the presence of a compensatory adaptive mechanism cannot be excluded in the improvement of balance function. At the same time, they became capable of raising the affected leg to the required height when stepping over obstacles. Results show that maintaining body balance is a complicated function, and that is why it is more difficult to recover in this respect. Interpreting the result, the restoration of lower limb functions after a stroke occurs faster than that of the upper limbs, fine motility, and body balance (example of trajectory, Figure 3c,d). Note that stroke rehabilitation in the context of the COVID-19 pandemic should be carried out considering the increased risks of intracerebral hemorrhage. This is because patients with COVID-19 infection have an increased risk of thrombotic events and receive various anticoagulation regimens, which increase the risk for intracerebral hemorrhage [49]. Therefore, motor rehabilitation should be carried out under the control of clinical and laboratory parameters with caution and proper dosing of aerobic motor training.

We also compared the results of our research with those of other studies using noninvasive engineering solutions in motor rehabilitation after stroke, such as a passive robotic device (Trackhold, PERCRO, Pisa, Italy), which works with five dedicated VR training applications for the upper limbs and is synchronized with a high-resolution electroencephalogram (HR-EEG) system [18]. As a result of using VR technology, motor recovery of the paretic upper extremities was observed, which resulted in increasing parameters reflecting the smoothness and accuracy of movements. In a clinical trial of VR training using the Xbox Kinect (XBOX 360, Kinect, Microsoft Inc), patients demonstrated clinical recovery of the upper extremities on the FMA-UE score [50]. In both studies, VR technologies were used in addition to conventional restorative therapy. In our study, motor rehabilitation using motion sensors and augmented reality was conducted as an independent rehabilitation program at the end of the early recovery period; hence, this allowed for us to separately evaluate the positive effects on motor-function recovery after stroke. The presented data of movement analysis, and the clinical and functional data, showed that AR rehabilitation proved to be effective as the poststroke rehabilitation method using the AR biofeedback motion training in our study. However, the results of our study are inconclusive with respect to the possible benefits of AR rehabilitation compared to those of classical rehabilitation techniques because the process of spontaneous recovery makes their benefits difficult to gauge. Furthermore, the physiological potential of neural plasticity for motor recovery may differ in study groups. Accordingly, we cannot say with certainty which rehabilitation program (one or more) would be sufficient for clinical degrees of motor impairment. If Group A patients had received only Phase I and AR rehabilitation, it is unlikely that they would have achieved the same degree of motor impairment recovery. Therefore, traditional and AR rehabilitation are not mutually exclusive but are complementary methods of rehabilitation in the early stroke recovery period. In addition, AR rehabilitation can be carried out ambulatory and can be useful for rehabilitation services in the context of the COVID-19 pandemic for reducing disability and optimizing functioning rehabilitation centers [51].

## 5. Limitations and Strengths

This study was carried out in patients with moderate and mild stroke; thus, stroke patients with severe disabilities, or people with severe cognitive or communication problems were excluded. Lack of randomization after the first stage of rehabilitation is also a disadvantage of this study. However, study results support the evidence for the existence of a special sensitive period for neuronal plasticity during the three months from stroke onset, showing the positive effect of AR rehabilitation in a relatively large sample. 

## 6. Conclusions

First, we note the importance of early rehabilitation, which was successful in patients with moderate ischemic stroke in our study. Second, our study showed that Phase I is not enough for the spontaneous recovery of motor impairment, and rehabilitation programs should be carried out throughout the entire sensitive period of poststroke plasticity. Summarizing, traditional exercise programs and AR rehabilitation have proven themselves in clinical practice. The provided data are convincing in that the newly developed AR biofeedback motion training is effective and safe as a separate rehabilitation method in the early recovery period of moderately severe, hemiparalytic, and ischemic stroke. These two rehabilitation approaches must be applied together or after each other, not instead of each other, as shown in clinical practice.

## Figures and Tables

**Figure 1 jcm-10-03718-f001:**
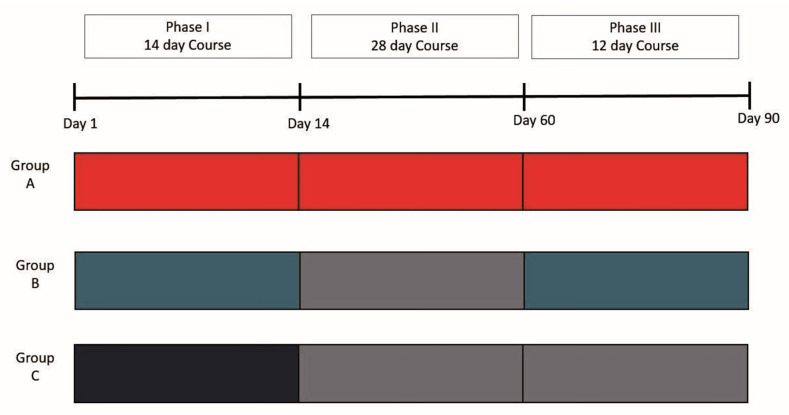
Study design.

**Figure 2 jcm-10-03718-f002:**
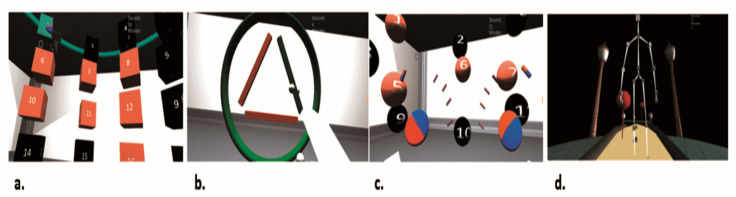
Visual interfaces for AR rehabilitation tasks: (**a**) Accuracy domain: motor domain focused on the movement accuracy of the upper limb; (**b**) Statics domain: motor domain focused on the static–dynamic load of muscles of the upper limb; (**c**) Capture domain: motor domain focused on the capture and compression of muscles of the hand; (**d**) Balance domain: motor domain focused on maintaining balance and overcoming obstacles when walking.

**Figure 3 jcm-10-03718-f003:**
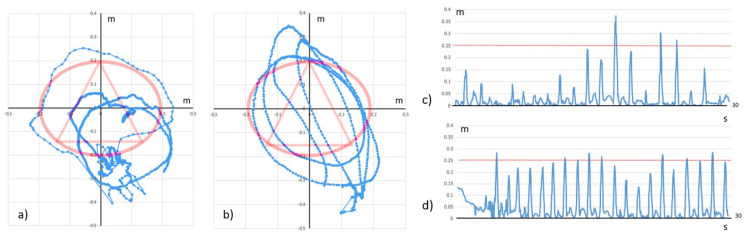
Trajectory of movement during AR rehabilitation on 1st and 10th days (Patient from A group example): (**a**) 1st session. Trajectory of movement on X and Y coordinates of forefinger on left hand during the Static domain execution: red line is given trajectory, blue line is real trajectory of forefinger; (**b**) 10th session. Trajectory of movement on X and Y coordinates of forefinger on left hand during the Static domain execution: red line is given trajectory, blue line is real trajectory of forefinger; (**c**) 1st session. Height of raising the left foot during the Balance domain execution: red line is height of virtual obstacle, blue line is Y coordinate of left foot; (**d**) 10th session. Height of raising the left foot during Balance domain execution: red line is height of virtual obstacle, blue line is Y coordinate of left foot.

**Table 1 jcm-10-03718-t001:** Baseline characteristics of the ischemic stroke patients.

	Ischemic Stroke *n* = 50			*p* _A–B_	*p* _B–C_	*p* _A–C_
	Group A *n* = 21	Group B *n* = 14	Group C *n* = 15			
Median Age, m (IQR), yr	62(57;67)	65(60;68)	66(60.5;68)	0.43	0.65	0.14
Sex				0.51	0.99	0.74
Male, *n* (%)	13(61.9%)	7(50.0%)	8(53.3%)			
Female, *n* (%)	8(38.1%)	7(50.0%)	7(46.7%)			
TOAST criteria, *n* (%)						
Large-artery atherosclerosis	2(9.5%)	3(21.4%)	2(13.3%)			
Cardioembolic	3(14.3%)	1(7.1%)	5(33.3%)			
Small artery occlusion	0(0.0%)	0(0.0%)	1(6.7%)			
Undetermined mechanism	15(71.4%)	8(57.1%)	6(40.0%)			
Other etiologies	1(4.8%)	0(0.0%)	0(0.0%)			
Hypertension, *n* (%)	21(100%)	14(100%)	15(100%)			
Duration of Hypertension m (IQR), yr	10(10;15)	14(10;15)	10(9;18)	0.94	0.86	0.83
Coronary heart disease, *n* (%)	5(23.8%)	3(21.4%)	7(46.7%)	0.99	0.25	0.18
Heart attack, *n* (%)	3(14.3%)	1(7.1%)	4(26.7%)	0.64	0.33	0.42
Atrial fibrillation, *n* (%)	3(14.3%)	4(28.6%)	4(26.7%)	0.4	0.99	0.42
Heart valve prosthesis, *n* (%)	0(0.0%)	0(0.0%)	2(13.3%)			
Dyslipidemia, *n* (%)	19(90.5%)	9(64.3%)	10(66.7%)	0.9	−0.99	0.1
Diabetes, *n* (%)	6(28.6%)	2(14.3%)	2(13.3%)	0.43	0.99	0.42
Duration of Diabetes m (IQR), yr	4(4;5)	8(5;10)	12(5;20)	0.14	0.66	0.14

Abbreviations: TOAST: Trial of Org 10172 in Acute Stroke Treatment; yr: years. Data are shown as median (m) and interquartile range (IQR), number (*n*) and percentage. * Estimated significant when *p* < 0.05.

**Table 2 jcm-10-03718-t002:** Estimated mean differences between patients groups.

	Group A	Group B	Group C	*p* _A–B_	*p* _B–C_	*p* _A–C_
	*n* = 21	*n* = 14	*n* = 15			
mRS-1	3(2;3)	3(2;4)	3(3;4)	0.86	0.85	0.68
mRS-2	3(2;3)	2(2;3)	2(2;3)	0.31	0.48	0.11
mRS-3	2(2;2)	2(2;3)	N/A	0.33	N/A	N/A
mRS-4	1(1;2)	1(1;2)	2(1;3)	0.93	0.01 *	0.05 *
NIHSS-1	5(3;6)	5(4;7)	6(3;8)	0.73	0.78	0.43
NIHSS-2	4(3;4)	4(3;4)	3(2;5)	0.93	0.72	0.61
NIHSS-3	3(2;4)	4(2;4)	N/A	0.61	N/A	N/A
NIHSS-4	2(1;3)	2(1;2)	3(2;5)	0.65	0.02 *	0.05 *
FMA-1	179(158;187)	187(178;194)	172(150;195)	0.28	0.45	0.95
FMA-shoulder-forearm-1	20(17;24)	23(18;26)	23(6;26)	0.33	0.78	0.63
FMA-wrist-hand-1	15(11;18)	17(10;20)	17(6;19)	0.52	0.78	0.78
FMA-UE-1	35(31;40)	39(28;45)	39(15;45)	0.5	0.72	0.9
FMA-LE-1	24(21;27)	26(22;28)	24(20;29)	0.28	0.51	0.99
FMA-Motor function (sum)-1	57(50;73)	67(55;73)	65(34;73)	0.45	0.72	0.95
FMA-Balance-1	10(9;12)	12(11;12)	11(5;13)	0.11	0.4	0.99
FMA-2	191(177;201)	206(194;212)	205(192;211)	0.01 *	0.72	0.12
FMA-shoulder-forearm-2	24(22;28)	29(28;32)	30(28;32)	0.02 *	0.88	0.08
FMA-wrist-hand-2	18(15;19)	23(19;25)	23(19;27)	0.00 *	0.75	0.01
FMA-UE-2	42(38;50)	53(49;54)	54(47;58)	0.01 *	0.72	0.02
FMA-LE-2	28(24;30)	31(26;34)	28(27;33)	0.08	0.53	0.36
FMA-Motor function (sum)-2	71(59;81)	82(80;87)	83(76;89)	0.01 *	0.95	0.04
FMA-Balance	12(10;13)	13(12;14)	12(12;14)	0.1	0.4	0.68
FMA-3	199(190;215)	207(195;217)	N/A	0.31	N/A	N/A
FMA-shoulder-forearm	27(25;34)	31(28;33)	N/A	0.33	N/A	N/A
FMA-wrist-hand	21(19;24)	24(20;25)	N/A	0.22	N/A	N/A
FMA-UE	49(43;57)	53(50;57)	N/A	0.33	N/A	N/A
FMA-LE	29(27;33)	32(29;34)	N/A	0.21	N/A	N/A
FMA-Motor function (sum)	78(71;89)	83(81;89)	N/A	0.26	N/A	N/A
FMA-Balance	12(12;14)	13(12;14)	N/A	0.47	N/A	N/A
FMA-4	213(208;222)	221(210;223)	205(192;213)	0.45	0.00 *	0.02 *
FMA-shoulder-forearm	33(30;36)	35(32;36)	30(28;32)	0.19	0.00 *	0.03 *
FMA-wrist-hand	27(25;28)	27(26;28)	24(19;27)	0.36	0.02 *	0.09 *
FMA-UE	61(56;64)	63(58;64)	54(47;59)	0.24	0.00 *	0.03 *
FMA-LE	33(29;34)	33(29;34)	29(27;33)	0.65	0.05 *	0.11
FMA-Motor function (sum)	93(84;97)	96(87;98)	84(76;91)	0.26	0.00 *	0.03 *
FMA-Balance	13(12;14)	14(13;14)	12(12;14)	0.41	0.06	0.2

Abbreviations: *n*: number of patients; N/A: not applicable; mRS: modified Rankin scale; NIHSS: national institutes of health stroke scale; FMA: Fugl-Meyer assessment; FMA-UE-upper extremity section of the Fugl-Meyer scale; FMA-LE—low extremity section of the Fugl-Meyer scale. Data are shown as median (m) and interquartile range (IQR). * Estimated significant when *p* < 0.05.

**Table 3 jcm-10-03718-t003:** Comparison between groups of patients who received rehabilitation in the early recovery period of ischemic stroke.

**Group A**								
***n* = 21**	**Point 1**	**Point 2**	**Point 3**	**Point 4**	***p*1–2**	***p*2–3**	***p*3–4**	***p*2–4**
mRS, m (IQR)	3(2;3)	3(2;3)	2(2;2)	1(1;2)	0.008 *	0.002 *	<0.001 *	<0.001 *
NIHSS, m (IQR)	5(3;6)	4(3;4)	3(2;4)	2(1;3)	0.001 *	0.009 *	<0.001 *	<0.001 *
FMA m (IQR)	179	191	199	213	<0.001 *	<0.001 *	<0.001 *	<0.001 *
	(158;187)	(177;201)	(190;215)	(201;222)				
FMA-shoulder-forearm	20(17;24)	24(22;28)	27(25;34)	33(30;36)	<0.001 *	<0.001 *	<0.001 *	<0.001 *
FMA-wrist-hand	15(11;18)	18(15;19)	21(19;24)	27(25;28)	<0.001 *	<0.001 *	<0.001 *	<0.001 *
FMA-UE	35(31;40)	42(38;50)	49(43;57)	61(56;64)	<0.001 *	<0.001 *	<0.001 *	<0.001 *
FMA-LE	24(21;27)	28(24;30)	29(27;33)	33(29;34)	0.002 *	0.001 *	0.001 *	<0.001 *
FMA-Motor function (sum)	57(50;73)	71(59;81)	78(71;89)	93(84;97)	<0.001 *	<0.001 *	<0.001 *	<0.001 *
FMA-Balance	10(9;12)	12(10;13)	12(12;14)	13(12;14)	0.003 *	0.06	0.002 *	<0.001 *
**Group B**								
***n* = 14**	**Point 1**	**Point 2**	**Point 3**	**Point 4**	***p*1–2**	***p*2–3**	***p*3–4**	***p*2–4**
mRS, m (IQR)	3(2;4)	2(2;3)	2(2;3)	1(1;2)	0.020 *	1	<0.001 *	0.001 *
NIHSS, m (IQR)	5(4;7)	4(3;4)	4(2;4)	2(1;2)	<0.001 *	0.010 *	<0.001 *	0.001 *
FMA, m (IQR)	187	206	207	221	<0.001 *	0.3	<0.001 *	0.001 *
	(178;194)	(194;211)	(195;213)	(210;223)				
FMA-shoulder-forearm	23(18;26)	29(28;32)	31(28;33)	35(32;36)	<0.001 *	0.11	<0.001 *	0.002 *
FMA-wrist-hand	17(10;20)	23(19;25)	24(20;25)	27(26;28)	<0.001 *	0.11	<0.001 *	0.001 *
FMA-UE	39(28;45)	53(49;54)	53(50;57)	63(58;64)	<0.001 *	0.11	<0.001 *	0.001 *
FMA-LE	26(22;28)	31(26;34)	32(29;34)	33(29;34)	0.010 *	0.07	0.08	0.038 *
FMA-Motor function (sum)	67(55;73)	82(80;87)	83(81;89)	96(87;98)	<0.001 *	0.07	<0.001 *	0.024 *
FMA-Balance	12(11;12)	13(12;14)	13(12;14)	14(13;14)	0.010 *	0.32	0.040 *	0.001 *
**Group C**								
***n* = 15**	**Point 1**	**Point 2**	**Point 3**	**Point 4**	***p*1–2**	***p*2–3**	***p*3–4**	***p*2–4**
mRS, m (IQR)	3(3;4)	2(2;3)	N/A	2(1;3)	<0.001 *	N/A	N/A	0.08
NIHSS, m (IQR)	6(3;9)	3(2;5)	N/A	3(2;5)	<0.001 *	N/A	N/A	0.1
FMA, m (IQR)	172	205	N/A	205	<0.001 *	N/A	N/A	0.75
	(149;194)	(192;210)		(192;213)				
FMA-shoulder-forearm	23(6;26)	30(28;32)	N/A	30(28;32)	<0.001 *	N/A	N/A	0.89
FMA-wrist-hand	17(6;19)	23(19;27)	N/A	24(19;27)	<0.001 *	N/A	N/A	0.85
FMA-UE	39(15;45)	54(47;58)	N/A	54(47;59)	<0.001 *	N/A	N/A	0.89
FMA-LE	24(20;29)	28(27;33)	N/A	29(27;33)	<0.001 *	N/A	N/A	0.79
FMA-Motor function (sum)	65(34;73)	83(76;89)	N/A	84(76;91)	<0.001 *	N/A	N/A	1
FMA-Balance	11(5;13)	12(12;14)	N/A	12(12;14)	0.010 *	N/A	N/A	0.18

Abbreviations: *n*: number of patients; N/A: not applicable; mRS: modified Rankin scale; NIHSS: national institutes of health stroke scale; FMA: Fugl-Meyer assessment; FMA-UE—upper extremity section of the Fugl-Meyer scale; FMA-LE—low extremity section of the Fugl-Meyer scale. Data are shown as median (m) and interquartile range (IQR). * Estimated significant when *p* < 0.05.

**Table 4 jcm-10-03718-t004:** Estimated mean differences in parameters of movement quality during AR rehabilitation between A and B groups.

	Group A	Group B	*p* _A–B_
	*n* = 21	*n* = 14	
Variability of movements, m (IQR),			
1st session	0.88(0.79,0.93)	0.89(0.86,0.9)	0.855
5th session	0.9(0.82,0.94)	0.89(0.86,0.92)	0.778
10th session	0.94(0.92,0.97)	0.95(0.91,0.97)	0.516
Number of completed tasks, m (IQR)			
1st session	21(19,25)	25(21,27)	0.118
5th session	36(31,46)	33(25,45)	0.495
10th session	56(47,64)	49(43,57)	0.396
Maximum duration in one approach (s), m (IQR)			
1st session	27(25,28)	23(18,26)	0.018 *
5th session	39(34,46)	38(34,44)	0.583
10th session	40(35,50)	45(37,50)	0.538
Variance of the displacement for the central point, m (IQR)			
1st session	0.08(0.08,0.09)	0.09(0.08,0.1)	0.554
5th session	0.07(0.08,0.09)	0.06(0.07,0.09)	0.712
10th session	0.04(0.03,0.06)	0.05(0.04,0.06)	0.528
Height of raising the paretic leg, m (IQR),			
1st session	0.22(0.12,0.35)	0.20(0.14,0.28)	0.282
5th session	0.34(0.32,0.40)	0.32(0.31,0.38)	0.343
10th session	0.36(0.34,0.40)	0.35(0.33,0.40)	0.532

Abbreviations: *n*: number of patients; s: seconds. Data are shown as median (m) and interquartile range (IQR). *Estimated significant when *p* < 0.05.

**Table 5 jcm-10-03718-t005:** Estimated mean differences in parameters of movement quality during AR rehabilitation in 1, 5, 10 sessions for A and B groups.

**Group A**						
***n* = 21**	**1st Session**	**5th Session**	**10th Session**	***p*1–5**	***p*5–10**	***p*1–10**
Variability of movements, m (IQR)	0.88(0.79;0.93)	0.90(0.82;0.94)	0.94(0.92;0.97)	<0.001 *	<0.001 *	<0.001 *
Number of completed tasks, m (IQR)	21(19;25)	36(31;46)	56(47;64)	0.001 *	<0.001 *	<0.001 *
Maximum duration in one approach (s), m (IQR)	27(25;28)	39(34;46)	40(35;50)	<0.001 *	0.434	<0.001 *
Variance of the displacement for the central point, m (IQR)	0.08(0.08:0.09)	0.07(0.08:0.09)	0.04(0.03:0.06)	0.144	0.001 *	<0.001 *
Height of raising the paretic leg, m (IQR)	0.22(0.12:0.35)	0.34(0.32:0.40)	0.36(0.34:0.40)	0.002 *	0.09	0.001 *
**Group B**						
***n* = 14**	**1st Session**	**5th Session**	**10th Session**	***p*1–5**	***p*5–10**	***p*1–10**
Variability of movements, m (IQR)	0.89(0.86;0.9)	0.89(0.86;0.92)	0.95(0.91;0.97)	0.022 *	0.001 *	<0.001 *
Number of completed tasks, m (IQR)	25(21;27)	33(25;45)	49(43;57)	0.016 *	0.001 *	<0.001 *
Maximum duration in one approach, (s), m (IQR)	23(18;26)	38(34;44)	45(37;50)	0.001 *	0.084	<0.001 *
Variance of the displacement for the central point, m (IQR)	0.09(0.08:0.1)	0.06(0.07:0.09)	0.05(0.04:0.06)	0.23	0.002 *	0.001 *
Height of raising the paretic leg, m (IQR)	0.20(0.14:0.28)	0.32(0.31:0.38)	0.35(0.33:0.40)	0.018 *	0.08	0.003 *

Abbreviations: *n*: number of patients; s: seconds. Data are shown as median (m) and interquartile range (IQR). * Estimated significant when *p* < 0.05.

## Data Availability

Data Availability Statements in section “MDPI Research Data Policies” at https://www.mdpi.com/ethics.

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
