# Peer review of "Clinical Evaluation of Different Treatment Strategies for Motor Recovery in Poststroke Rehabilitation during the First 90 Days"

_jcm, 2021, doi:10.3390/jcm10163718_

Round 1
Reviewer 1 Report
Interesting and well written study regarding motor plasticity in post stroke phase.
Thanks for the opportunity to review this paper.
The study highlights a concept which is Usually underestimated in common clinical practice.
I would just suggest to add to add this reference: Pilato et al. Predicting factors of functional outcome in patients with acute ischemic stroke admitted to neuro-intensive care unit. A prospective cohort study. Brain sciences 2020, 10(12):1-14, 911.
To underline and better clarify the difference in acute stroke care inside the hospital.
Author Response
Reviewer 1.
Interesting and well written study regarding motor plasticity in post stroke phase.
Thanks for the opportunity to review this paper.
The study highlights a concept which is Usually underestimated in common clinical practice.
I would just suggest to add this reference: Pilato et al. Predicting factors of functional outcome in patients with acute ischemic stroke admitted to neuro-intensive care unit. A prospective cohort study. Brain sciences 2020, 10(12):1-14, 911.
To underline and better clarify the difference in acute stroke care inside the hospital.
Response: We thank Reviewer #1 for his/her such a high assessment of the article and suggestions.
We added the reference: Pilato et al. Predicting factors of functional outcome in patients with acute ischemic stroke admitted to neuro-intensive care unit. A prospective cohort study. Brain sciences 2020, 10(12):1-14, 911.
- Discussion
… Secondary safety outcomes were the absence of mortality and intracranial hemorrhages in patients who had received early rehabilitation. The intensity of rehabilitation was moderate, and the modes of early rehabilitation were selected individually for each patient. The professionals most likely applied additional predictors of successful recovery, helping to avoid complications and clinical deterioration. Specific clinical factors of patients with acute ischemic stroke affect not only the outcome of early rehabilitation. They are also considered when using other treatment approaches for acute ischemic stroke. The recent prospective cohort study was shown that intra-arterial mechanical thrombectomy either in combination with intravenous thrombolysis or alone in patients with acute ischemic stroke can be associated with good outcomes at 3 months (Pilato et al.,2020). The determinant for 90-day were a baseline NIHSS score. It is also shown that the reperfusion seems to be not the only determinant of good outcomes. It can be supposed that some aspects of patient management in the intensive care unit, individual therapeutic and rehabilitation approaches in acute phase of ischemic stroke can counterbalance the positive effects of reperfusion and further improve neurological outcomes.

Reviewer 2 Report
- Introduction is very good, but the aim of this paper is not highlighted enough at this point (lines 113-117), Please improve.
- It is not clear how those patients were divided into 3 groups (A, B, C). Randomness? In chronological order?
- How long was follow-up of those patients?
- Although not the topic of this paper, the importance of rehabilitation during the Covid-19 pandemic must be shown, just a few lines at these point (lines 551-570). Please look at these 2 refs: Intracranial hemorrhage and COVID-19, but please do not forget "old diseases" and elective surgery. Brain Behav Immun. 2021 Feb;92:207-208. 10.1016/j.bbi.2020.11.034. ---- COVID-19 pandemic. What should Physical and Rehabilitation Medicine specialists do? A clinician's perspective. Eur J Phys Rehabil Med. 2020 Aug;56(4):515-524. doi: 10.23736/S1973-9087.20.06317-0.
- "This study was carried out in patients with moderate and mild stroke; thus, stroke patients with severe disabilities, or people with severe cognitive or communication prob lems were excluded" (lines 601-603). This must also be stated in the materials and methods section.
- "Patents. This section is not mandatory but may be added if there are patents resulting from the work reported in this manuscript" (618-620) Authors forgot to remove this sentence.
- A revision is needed.
Author Response
We thank Reviewer #2 for their positive evaluation of our study and helpful criticisms and suggestions, following which we significantly modified our manuscript. We believe that these changes have significantly improved our manuscript and clarified our data presentation. Below, please find our response to specific comments made by Reviewer #2.
Reviewer 2.
- Introduction is very good, but the aim of this paper is not highlighted enough at this point (lines 113-117), Please improve.
Response: We thank the reviewer for pointing this out. We changed the aim:
The aim of this study was to evaluate the role of different rehabilitation strategies and their combinations for motor recovery and decreasing of functional disability as of neurological and functional outcomes at 3 months of ischemic stroke.
- It is not clear how those patients were divided into 3 groups (A, B, C). Randomness? In chronological order?
Response: We thank the reviewer for this comment.
Patients were divided into 3 groups (A, B, C) not randomness.
The absence of treatment in the II and III phases of rehabilitation in group C and in the phase II of rehabilitation in group B was associated with problems of routing and optimizing Phases II and III of the rehabilitation process for cerebral stroke and problems in the family. Most of these patients lived in geographically remote areas of the Tomsk region, did not have the opportunity to stay in the regional center and pass Phase II of rehabilitation after treatment in Tomsk RSC, where they were taken by air ambulances within the first 24 hours of stroke.
- How long was follow-up of those patients?
Response: We thank the reviewer for this comment.
The study lasted 90 days. Patients were followed up on an outpatient care after the end of the study, with a frequency of 1 time per month.
- Although not the topic of this paper, the importance of rehabilitation during the Covid-19 pandemic must be shown, just a few lines at these point (lines 551-570). Please look at these 2 refs: Intracranial hemorrhage and COVID-19, but please do not forget "old diseases" and elective surgery. Brain Behav Immun. 2021 Feb;92:207-208. 10.1016/j.bbi.2020.11.034. ---- COVID-19 pandemic. What should Physical and Rehabilitation Medicine specialists do? A clinician's perspective. Eur J Phys Rehabil Med. 2020 Aug;56(4):515-524. doi: 10.23736/S1973-9087.20.06317-0.
Response: We thank the reviewer for this comment. This is an important subject clearly. We added a few lines and referred the article data:
….. Note that stroke rehabilitation in the context of COVID-19 pandemic should be carried out considering the increased risks of intracerebral hemorrhage. This is because patients with COVID-19 infection have an increased risk of thrombotic events and receive various anticoagulation regimens, which increase the risk for intracerebral hemorrhage (Montemurro, 2021). Therefore, motor rehabilitation should be carried out under the control of clinical and laboratory parameters with caution and proper dosing of aerobic motor training. (line 570)
… In addition, AR rehabilitation can be carried out ambulatory and can be useful for rehabilitation services in the context of COVID-19 pandemic for reduce disability and optimize the functioning rehabilitation centers (Carda et al., 2020). (line 599)
- "This study was carried out in patients with moderate and mild stroke; thus, stroke patients with severe disabilities, or people with severe cognitive or communication problems were excluded" (lines 601-603). This must also be stated in the materials and methods section.
Response: We thank the reviewer for this comment. We absolutely agree and added it:
2.2 Patient Characteristics
Fifty stroke patients aged 18 years or more were recruited from the Tomsk RSC within the first 24 h after the onset of symptoms. Ischemic stroke in the middle cerebral artery basin was diagnosed considering their clinical history, the presence of focal neurological signs and/or symptoms, and by applying magnetic resonance imaging (MRI) or computer tomography (CT) scans. Additional inclusion criteria were Glasgow Coma Scale (GCS); of no less than 15 points, and Mini-Mental State Examination (MMSE); of no less than 20 points (Teasdale et al., 2014; Teasdale & Jennett, 1974; Folstein et al., 1975). Thus, this study was carried out in patients with moderate and mild stroke. Stroke patients with severe disabilities, or people with severe cognitive or communication problems were excluded. Excluded were patients with subarachnoid, extradural, or subdural hemorrhage; transient ischemic attack; history of prior stroke; neurological deficits related to trauma, neoplasm, or psychiatric disorders; and those with excessive use of alcohol or drugs. Patients were also excluded who had received thrombolysis and thrombectomy, who had developed hemorrhagic transformation, or who were clinically unstable.
- "Patents. This section is not mandatory but may be added if there are patents resulting from the work reported in this manuscript" (618-620) Authors forgot to remove this sentence.
Response: We thank the reviewer for this comment. We want to remove this sentence.
- A revision is needed.
Response: We have changed manuscript according to reviewer’s comments.

Round 2
Reviewer 2 Report
Authors well revised the manuscript.